# A pilot study of inflammatory mediators in brain extracellular fluid in paediatric TBM

Nicholas W. Loxton[1], Ursula K. Rohlwink[1,2,3], Mvuwo Tshavhungwe[1], Lindizwe Dlamini[1], Muki Shey[4,5], Nico Enslin[1,2], Anthony Figaji[1,2]*

1 Division of Neurosurgery, Department of Surgery, University of Cape Town, Cape Town, South Africa,
2 Neuroscience Institute, University of Cape Town, Cape Town, South Africa, 3 The Francis Crick Institute, London, United Kingdom, 4 Wellcome Centre for Infectious Disease Research in Africa, Cape Town, South Africa, 5 Department of Medicine, University of Cape Town, Cape Town, South Africa

* Anthony.Figaji@uct.ac.za

## Abstract

Tuberculous meningitis (TBM) is the most fatal form of tuberculosis and frequently occurs in children. The inflammatory process initiates secondary brain injury processes that lead to death and disability. Much remains unknown about this cerebral inflammatory process, largely because of the difficulty in studying the brain. To date, studies have typically examined samples from sites distal to the site of disease, such as spinal cerebrospinal fluid (CSF) and blood. In this pilot study, we examined the feasibility of using direct brain microdialysis (MD) to detect inflammatory mediators in brain extracellular fluid (ECF) in TBM. MD was used to help guide neurocritical care in 7 comatose children with TBM by monitoring brain chemistry for up to 4 days. Remnant ECF fluid was stored for offline analysis. Samples of ventricular CSF, lumbar CSF and blood were collected at clinically indicated procedures for comparison. Inflammatory mediators were quantified using multiplex technology. All inflammatory markers, with the exception of interleukin (IL)-10 and IL-12p40, were detected in the ECF. Cytokine concentrations were generally lower in ECF than ventricular CSF in time-linked specimens. Individual cases showed ECF cytokine increases coinciding with marked increases in ECF glycerol or decreases in ECF glucose. Cytokine levels and glycerol were generally higher in patients with more severe disease. This is the first report of inflammatory marker analysis from samples derived directly from the brain and in high temporal resolution, demonstrating feasibility of cerebral MD to explore disease progression and possibly therapy response in TBM.

## Introduction

Tuberculous meningitis (TBM) is an often-lethal manifestation of *M. tuberculosis* infection. Neurological disability and developmental deficit in young children is common in survivors [1]. The poor outcome is usually due to the consequences of brain inflammation, which primarily manifests as an inflammatory infiltrate in the basal subarachnoid space of the brain, causing arachnoiditis, obliterative vasculitis, variable encephalitis, and hydrocephalus [2].

**Data Availability Statement:** All relevant data are uploaded to the University of Cape Town's Open Data repository ZivaHub and publicly accessible via

the following URL: https://doi.org/10.25375/uct.13807943.v1.

**Funding:** The study was supported by the National Research Foundation SARChI Chair of Clinical Neurosciences, Harry Crossley Foundation and South African Medical Research Council, National Medical Students Research Training Programme received by Nicholas Loxton. The funders had no role in study design, data collection and analysis, decision to publish, or preparation of the manuscript.

**Competing interests:** The authors have declared that no competing interests exist.

These eventually lead to raised intracranial pressure and brain ischaemia, which lead to poor outcomes. Understanding the neuroinflammatory response in TBM is thus critical in developing novel host-directed therapies.

Concentrations of inflammatory markers are higher in the cerebrospinal fluid (CSF) than the blood in TBM patients, suggesting compartmentalisation of the immune response at the site of disease in the central nervous system (CNS) [3, 4], although some of the cytokine burden may arise from peripheral sources. Cytokine concentrations are higher in spinal (lumbar) than ventricular CSF (VCSF), possibly due to contributions from spinal pathology and relative stagnation of CSF flow in the spinal subarachnoid space, which increases contributions from systemic sources [5–7]. Therefore, to understand the inflammatory process occurring in the brain itself, it is preferable to sample as close to brain tissue as possible. This is rarely possible ante-mortem and single samples do not reflect dynamic processes in the brain—temporal profiling has to date been impossible. VCSF may be more informative but high frequency repeated sampling cannot be done without breaking the sterility seal of the ventricular drainage system.

Cerebral microdialysis (MD) may offer a unique opportunity to sample the brain extracellular fluid (ECF) directly and frequently. It involves placement of a semi-permeable dialysis catheter directly in brain tissue, which enables hourly sampling of substances in the ECF over 4–5 days. This may enable high temporal resolution examination of the inflammatory processes occurring at the site of disease during the acute phase of illness.

MD monitoring has been used for various acute brain conditions requiring intensive care unit (ICU) management [8, 9]. This dynamic sampling of the brain ECF may provide a more biologically relevant picture than other fluid compartments as it measures the molecules at their sites of action and secretion [9]. Typical bedside use of MD for clinical purposes is performed by monitoring metabolic markers of energy crisis (glucose, lactate, pyruvate, glutamate) and cellular injury (glycerol) and titrating interventions based on their changes. More recently, inflammatory markers have been examined in traumatic brain injury and have been shown to be associated with outcome [10–12]. Although MD has been used in several different forms of neuroinflammation [10, 13], it has not yet been applied in TBM. This pilot study aimed to investigate the feasibility of using MD to detect cytokines in brain ECF of patients with TBM.

## Methods and materials

Patients presenting to the Red Cross War Memorial Children's Hospital with definite or probable TBM [14] who required ICU admission with ventilation and intracranial monitoring were eligible. Demographic and clinical data were collected. Patients were managed as per hospital protocol, described elsewhere [15]. MD was used as part of clinical bedside brain monitoring of brain metabolites. Remnant fluid, after bedside clinical analysis was performed, was stored for research purposes. Other routine brain monitoring included brain physiology (brain oxygenation and intracranial pressure), management of which has been previously described [16].

We reviewed the radiological findings from formal reports of brain computed tomography (CT) scans, documenting the presence and extent of infarction. We classified patients into two groups based on the characteristics of infarcts: Group A comprised patients with either focal infarcts or no infarcts; Group B comprised patients with extensive hemispheric or bilateral infarcts. We examined associations between cytokines and metabolites (lactate-pyruvate ratio, glycerol and glucose) with the two radiological groups and mortality at one year.

This study was approved by the University of Cape Town Human Research Ethics Committee (HREC 564/2012). Written informed consent was obtained from patients' legal guardians.

## Sample collection

The MD catheter, a thin, dual lumen tube with a distal semi-permeable membrane with 100kDa pore size (MDialysis, Stockholm, Sweden), was inserted into right frontal white matter. A perfusion fluid isotonic with CSF (CNS Perfusion fluid, MDialysis) was pumped at 0.3μL/min through the catheter, allowing molecules in the brain ECF to diffuse into the MD perfusate along a concentration gradient [17]. The perfusate was collected and analysed hourly at the bedside for glucose, glutamate, lactate, pyruvate and glycerol (metabolites) as part of clinical care. Remnant fluid was kept on ice and biobanked at -80˚C within 6 hours of collection. As MD sample volumes are typically small, hourly samples were pooled over 4-hour epochs to ensure sufficient volume for Luminex Multiplex assay analysis (>25μL). The first 2 hours of sample after the MD catheter placement was not analysed to avoid possible insertion artefacts. We examined a daily ECF sample (usually 3am to 7am) for up to 4 days from admission (the maximum duration of MD monitoring). For time-linked comparisons, we also analysed ECF samples collected at the same time as other clinically indicated procedures that sampled VCSF, lumbar CSF (LCSF) and plasma. These included blood samples, CSF sampling from an external ventricular drainage system, and/or lumbar puncture. In some patients, column tests or air encephalograms were performed to determine the level of CSF obstruction causing hydrocephalus, as per our institutional protocol [15]. This enabled simultaneous sampling of LCSF and VCSF.

## Sample analysis

Samples were analysed for Growth-Regulated Oncogene (GRO), Interferon-γ (IFN-γ), Interferon-γ inducible protein-10 (IP-10), Interleukin (IL) -1β, IL-1 Receptor Antagonist (IL-1Ra), IL-6, IL-8, IL-10, IL-12p40, Monocyte Chemoattractant Protein 1 (MCP-1), Macrophage Inflammatory Protein -1α (MIP-1α), Tumour Necrosis Factor-α (TNF-α) and Vascular Endothelial Growth Factor (VEGF) using Luminex technology on the Bio-Plex platform (Bio-Rad Laboratories, Hercules, CA, USA). We used a custom 13 Plex MILLIPLEX Map Human Cytokine / Chemokine Magnetic Bead Panel Kit (EMD Millipore Corporation, Billerica, MA, USA) as per the manufacturer's instructions with the serum matrix as previously validated within our lab.

## Statistical analysis

Data are presented as median, quartiles, minimum and maximum values over the first 4 days of monitoring. Concentrations below the lower limit of detection for the kit were recorded as 0.01pg/mL for analysis. The R statistical package [18] was used to analyse data and generate figures. Due to the small cohort size and the large number of repeated measures, we did not perform statistical analysis of the relationship with radiology and clinical outcome. Rather, we highlighted variables where there was at least a 3-fold difference in cytokines and metabolites between the groups to generate hypotheses.

# Results

## Demographics and outcomes

Seven patients with a median age of 2.3 (range 0.6–12.6) years were enrolled. Definite TBM was diagnosed by CSF culture in 1 patient, the remaining 6 patients met the definition of probable TBM according to the published case definition [19]. All patients presented in British Medical Research Council TBM Stage 3 [20]; all were comatose and ventilated in the ICU. None was HIV co-infected. Three patients died. Few patients had LCSF samples available for

analysis in the first few days of hospitalisation, therefore, VCSF measurements are reported (all patients had external ventricular drainage for hydrocephalus). VCSF is known to be different from LCSF, the latter of which is more commonly reported for TBM [3]. Brain computed tomography (CT) scans showed typical features, namely hydrocephalus, basal meningeal enhancement, and infarcts in all patients (n = 7). Two patients had clinically indicated spinal magnetic resonance imaging, which demonstrated spinal disease in both. Four patients showed features of pulmonary TB on chest radiographs. Demographic-, clinical admission features, admission chemistry- and imaging- characteristics are summarised in Table 1.

**Table 1. Demographic, admission clinical, admission chemistry and imaging characteristics.**

| Characteristic | Value |
|---|---|
| Demographic characteristics | |
| Age, years | 2.3 (0.7–12.6) |
| Sex | |
| Male | 5 (71) |
| Clinical characteristics | |
| Fever | 4 (57) |
| Altered level of consciousness | 7 (100) |
| Focal neurology | 6 (86) |
| Seizures | 5 (71) |
| Meningism | 1 (14) |
| HIV infection | 0 (0) |
| Immunisations up to date | 3 (43) |
| Surgery performed | |
| EVD insertion | 7 (100) |
| VPS insertion | 3 (43) |
| Diagnostics | |
| CSF Culture Positive | 1 (14) |
| Outcome | |
| Mortality | 3 (43) |
| VCSF Chemistry | |
| Glucose (mmol/L) | 3.7 (1.5–4.6) |
| Chloride (mmol/L) | 112 (93–124) |
| Protein (g/L) | 0.57 (0.27–1.35) |
| VCSF Cells | |
| Polymorphonucleocytes (cells/μL) | 6 (0–37) |
| Lymphocytes (cells/μL) | 18 (6–45) |
| WCC (cells/μL) | 38 (7–58) |
| Imaging Data | |
| Hydrocephalus | 7(100) |
| Enhancement | 7(100) |
| Tuberculoma | 0(0) |
| Spinal Disease (n = 2) | 2 (100) |
| Infarcts | 7 (100) |
| Chest X-ray suggestive of PTB | 4 (57) |

Data presented as median (range) or n (%). Imaging data from all in-hospital scans. Only 2 patients were assessed for spinal disease on imaging. VCSF obtained on admission. Abbreviations: EVD, external ventricular drain; VPS, ventriculoperitoneal shunt; VCSF, ventricular cerebrospinal fluid; WCC, White Cell Count.

## Inflammatory markers in ECF

Summaries of the ECF data are presented in S1 Table and Fig 1. As per clinical indications for MD, which determined duration of monitoring, ECF samples were available for 5 epochs (Admission; Day 1–4) in 6 patients, and only 3 epochs (Admission; Day 1–2) in patient 7.

Inflammatory markers GRO, IFN-γ, IP-10, IL-1β, IL-1Ra, IL-6, IL-8, MCP-1, MIP-1α, TNF-α and VEGF were all detected in at least 1 time point in the ECF overall. IL-10 and IL-12p40 were below the kit's lowest detectable limit for all time points. Relative to the other analytes, MCP-1 consistently showed the highest median concentrations in ECF at each time-point, IP-10 showed the second highest concentrations for the first 4 time points, and IL-1Ra's median concentration was highest on the final day of monitoring (Day 4). Median GRO, MIP-1α and TNF-α was usually below the detection limit for all days analysed. Temporal profiles varied across patients as demonstrated for MCP-1 in Fig 2.

There were no obvious differences in the ECF concentrations of patients who survived relative to those who died. However, this study was not powered for an outcome analysis.

## Examination of cytokine spikes with ECF metabolic data for individual patients

Time-linked spikes in cytokine concentrations were seen in several patients. To explore the relationship between cytokines and other parameters we examined the time-linked inflammatory, metabolic, and physiological data in individual patients. In patient 2, the spike seen in several cytokines including IL-6 and MIP-1α on day 1 of monitoring occurred simultaneously with a sharp rise in ECF glycerol, a marker of cell membrane breakdown (Fig 3A). Progressive or secondary increases in glycerol may denote ongoing or evolving secondary injury. For patient 6, a spike in inflammatory markers GRO and IL-8 on day 3 coincided with a steady increase in ECF glycerol and a rise in the lactate/pyruvate ratio (from 20 to 40) with critically low brain oxygen levels (<10 mmHg), both of which were in keeping with brain ischaemia

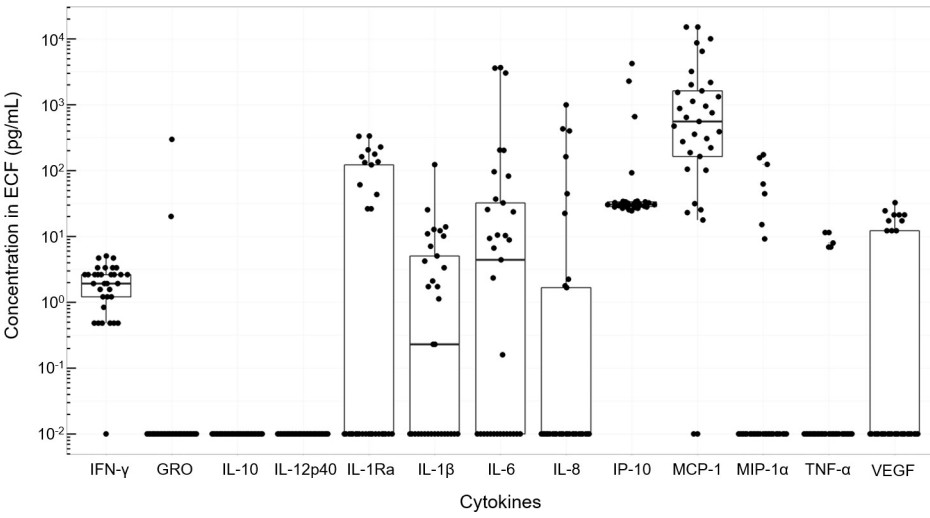

**Fig 1. Distributions of inflammatory marker concentrations recovered from brain extracellular fluid.** All microdialysis concentration values plotted for each of the 13 markers (n = 33) overall (not parsed by time point or by patient). Points plotted in a quasi-random fashion horizontally to avoid over-plotting. Abbreviations: ECF, extracellular fluid; GRO, growth-regulated oncogene; IFN-γ, interferon-γ; IL, interleukin; IL-1Ra, interleukin-1 receptor antagonist; IP-10, interferon-γ inducible protein-10; MCP-1, monocyte chemoattractant protein; MIP-1α, macrophage inflammatory protein 1α; TNF-α, tumour necrosis factor-α; VEGF, vascular endothelial growth factor.

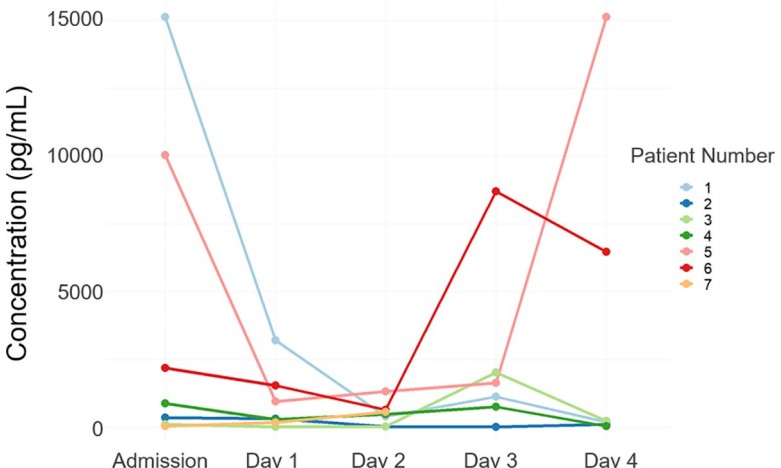

**Fig 2. Temporal concentrations of inflammatory markers recovered through microdialysis monitoring.**
Concentrations (pg/mL) of inflammatory marker MCP-1 recovered from brain ECF obtained through microdialysis monitoring. Patient numbers were arbitrarily assigned. Patient lines are not visible where another is plotted directly on top of it (e.g. both patient's values remain at baseline). Abbreviations: ECF, extracellular fluid; MCP-1, monocyte chemoattractant protein.

(Fig 3B). In Patient 5 a spike in inflammatory markers on the final day of monitoring (Day 4) coincided with the lowest brain glucose concentrations for the monitoring period and intermittent decreases in brain oxygenation (Fig 3C).

## Recovery of cytokines in different fluid compartments

VCSF samples were the second most abundant (n = 9) in our dataset and were available for 5 of our patients. Time-linked sample analysis showed lower concentrations in ECF than the VCSF in general (Fig 4). This difference was less clear for IL-1β, MCP-1, MIP-1α, and IL-1Ra, which displayed concentrations that were comparable or even higher than their paired VCSF samples in some patients. In patients who also had time-linked LCSF samples (n = 3) recorded, concentrations tended towards being highest in the LCSF, followed by VCSF, and then ECF (S1 Fig). S2 Fig summarises how the plasma samples compare to their corresponding VCSF and ECF samples, where available (n = 3).

## Associations of MD cytokines and metabolites with radiological and clinical outcome

There were 33 cytokine samples in total and 459 metabolite samples in 7 patients. Two patients demonstrated extensive or bilateral infarction (Group B); 5 patients demonstrated focal [n = 4] or no infarction [n = 1] (Group A). Three patients had died at one year follow up; 4 survivors all achieved a good functional outcome. The relevant values are shown in S2 and S3 Tables.

**Radiology.** For cytokines:

1. Median cytokine concentrations in Group B were greater than or equal to corresponding values in Group A for all cytokines.

2. Cytokines that demonstrated median concentrations at least 3-fold greater: IL-1β, IL-1Ra, IL-6, and MCP-1 (in Group B).

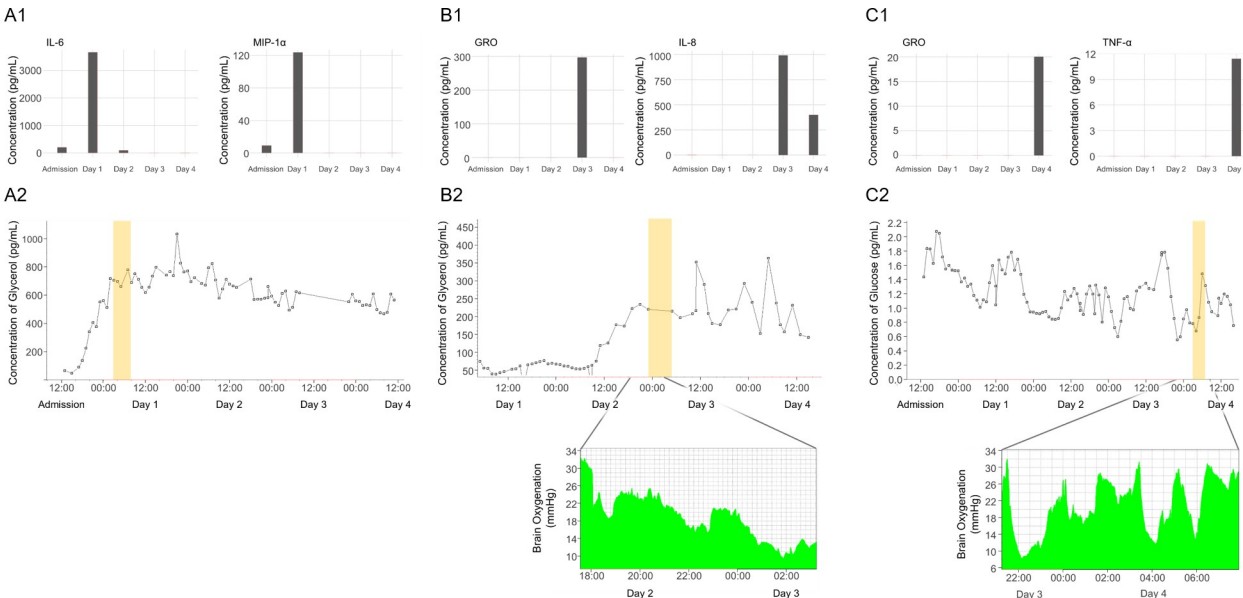

**Fig 3. Inflammatory marker spikes time-linked with metabolic data.** (A) Patient 2. (A1) Representative inflammatory markers illustrating a temporally linked spike in MIP-1α and IL-6 concentrations on Day 1 in the ECF. (A2) Glycerol concentrations in the ECF of patient 2 which show a peak without a rapid subsequent decline, indicating ongoing injury. (Glycerol is a marker of cellular injury). The time-period during which the Day 1 ECF sample was obtained is shown by the yellow rectangle. (B) Patient 6. (B1) Representative inflammatory markers showing temporally linked spike in GRO and IL-8 concentrations on Day 3 in the ECF. (B2) Upper panel: Glycerol concentrations in the ECF of patient 6. The time-period during which the Day 3 ECF sample was obtained is shown by the yellow rectangle and follows a rise in glycerol. Glycerol levels do not decrease thereafter, indicating ongoing injury. The lower panel shows the brain tissue oxygen monitoring including the time period during which the sample was taken (normal brain oxygen is greater than 20mmHg; 10mmHg is a critical threshold for brain hypoxia). (C) Patient 5. (C1) Representative inflammatory markers showing temporally linked spike in GRO and TNF-α concentrations on Day 4 in the ECF. (C2) Upper panel: Glucose concentrations in the ECF of patient 5. The time-period during which the Day 4 ECF sample was obtained is shown by the yellow rectangle and is associated with the low glucose concentration. The lower panel shows the corresponding fluctuation in the brain tissue oxygen monitoring including the time period during which the sample was taken (normal brain oxygen is greater than 20mmHg; 10mmHg is a critical threshold for brain hypoxia). Abbreviations: ECF, extracellular fluid; GRO, growth-regulated oncogene; IL-6, interleukin-6; IL-8, interleukin-8; MIP-1α, macrophage inflammatory protein 1α; TNF-α, tumour necrosis factor-α.

**For metabolites:.**  Median glycerol concentrations were 3-fold greater in Group B; the lactate-pyruvate ratio was slightly higher and glucose was slightly lower.

**Clinical outcome.**  For cytokines:

1. Median cytokine concentrations in patients who died were greater than or equal to corresponding values in survivors for all cytokines, except IL-6.

2. Cytokines that demonstrated median concentrations at least 3-fold greater: IL-1β and MCP-1 (in patients who died).

   For metabolites:
   Median glycerol concentrations were 3-fold greater in patients who died; the lactate-pyruvate ratio was slightly higher, and glucose was slightly lower.

## Discussion

This pilot study demonstrated the feasibility of 1) detecting a range of inflammatory markers in brain ECF of paediatric TBM patients using MD monitoring, and 2) describing temporal changes in ECF cytokine concentrations over several days. These patterns may associate with the physiological state of the patient, as manifest by corresponding changes in brain chemistry and brain oxygenation. However, the sample size is small, and this requires validation in a

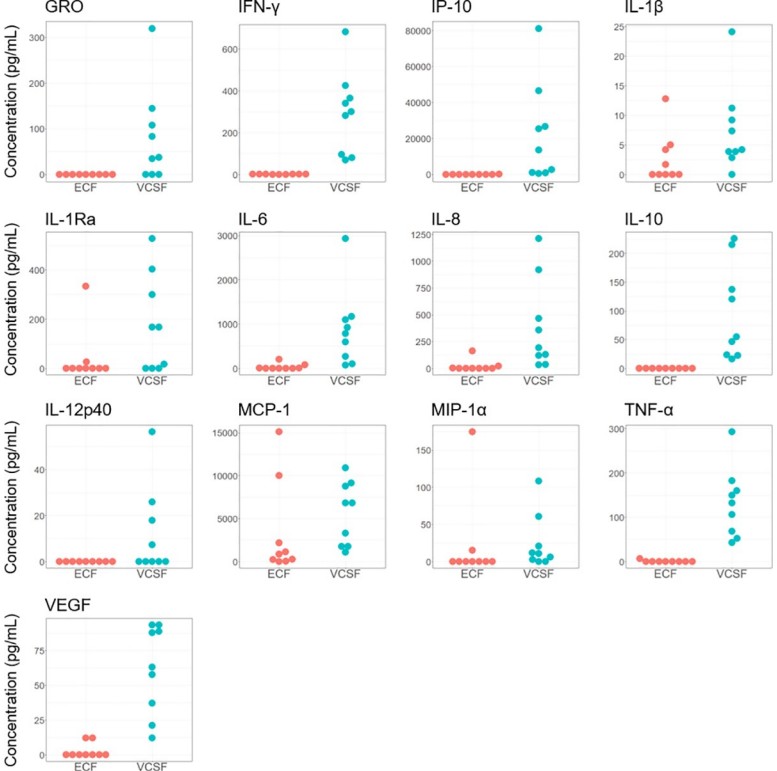

**Fig 4. Paired brain extracellular and ventricular cerebrospinal fluid cytokine concentrations.** Each VCSF sample analysed (n = 9) is plotted with its paired ECF which was obtained from the same patient at the same time point. Plotted values have been shifted horizontally to avoid over-plotting. Abbreviations: ECF, extracellular fluid; GRO, growth-regulated oncogene; IFN-γ, interferon-γ; IL, interleukin; IL-1Ra, interleukin 1 receptor antagonist; IP-10, interferon-γ inducible peptide; MCP-1, monocyte chemoattractant protein; MIP-1α, macrophage inflammatory protein 1α; TNF-α, tumour necrosis factor-α; VEGF, vascular endothelial growth factor; VCSF, ventricular cerebrospinal fluid.

larger cohort. Concentrations in samples may be affected by relative recovery rates and anatomical location of catheter placement; however, it offers a potentially useful method for tracking changes over time and in response to conventional or novel therapies.

## Recovery of inflammatory markers using cerebral microdialysis

This is the first time that molecules in brain ECF were examined in TBM. Cytokines were in some cases below the lower limit of detection. This mirrors what is seen in traumatic brain injury in a paper by *Helmy et al.* [11], whose examination included the same cytokines in our study and which similarly showed MCP-1 and IP-10 to be the highest cytokines.

Several factors might affect the recovery of molecules from brain ECF, including placement of the catheter, the relative recovery for a particular molecule, the type of perfusion fluid and flow rate, insertion artefact, occasional local haemorrhage around the probe, and bio-fouling of the catheter by brain particles and blood [11, 21–23]. However, in this cohort the MD catheters were placed under direct vision, subsequent imaging confirmed correct position, initial ECF samples were discarded and there were no focal haemorrhages. The low analyte concentrations in our data could, therefore reflect a) that the molecule was not present in the ECF, although this is less likely given that those analytes were present in the VCSF, b) it was not detected due to insufficient sensitivity of the analysis kit for low concentrations or c) the

relative recovery of the molecule through the microdialysis testing modality was insufficient to detect low concentrations. The use of dextran or albumin as a perfusion fluid may have increased the yield [24, 25]. It is promising, however, to be able to generate pilot data on ECF cytokine concentrations in TBM.

## Temporal profile

The temporal profiles over the course of monitoring were mixed across patients. This contrasts to what has been found in traumatic brain injury [11], and may reflect different pathophysiological dynamics; however, the sample size of this study was small.

It is noteworthy that the spikes observed in the ECF cytokine concentrations coincided with changes in ECF glycerol, ECF glucose, lactate-pyruvate ratio, or brain oxygen. Glycerol is a component of the cell membrane and elevated levels are considered to be associated with brain cell damage and membrane breakdown that may be due to ischaemia [26] and tissue injury [27]. This may also manifest as low brain oxygenation and an increased lactate/pyruvate ratio (a marker of perturbed glucose metabolism). Ischaemia occurs commonly in TBM, often even after starting full therapy, and drives poor patient outcomes, but we still do not have a full understanding of what causes this progression and how it evolves in individual patients. These data may offer insight into the disease processes occurring, and the immediacy of the negative impact on the brain. It may also help determine the cerebral response of the disease to novel therapies.

## Compartmental differences

The compartmental comparisons between the VCSF and ECF (n = 9 paired samples) yielded interesting differences. Most cytokines showed higher concentrations in the VCSF, which may reflect true differences [9] or the reduced relative recovery through the MD catheter. The relative variation between ratios of different cytokines, though, suggest that at least some compartmental effects exist, given the static physico-chemical properties of the MD membrane. It is worth noting that MIP-1α and MCP-1 showed relatively high concentrations when compared to the VCSF compartment, potentially suggesting greater expression in the brain. Further interpretation of this is not possible due to the small number of paired samples. The differences in cytokine concentrations between time-linked LCSF and VCSF samples also support our previous work demonstrating CNS compartmental differences [3].

## Associations with clinical and radiological outcome

Although the sample size is small, the data suggest some intriguing hypotheses that require examination in a larger cohort. In general, cytokine values were greater in patients who died, and who experienced extensive infarction. In particular, IL-1β, IL-1Ra, IL-6, and MCP-1 showed 3-fold higher values in patients with extensive infarction; for patients who died, IL-1β was 3-fold greater and MCP-1 showed just less than a 3-fold difference (S2 Table).

Metabolites showed less marked differences, but glycerol was 3-fold greater in patients who died and patients who developed extensive or bilateral infarction, which may be in keeping with tissue destruction (S3 Table).

These observations require validation in a larger cohort, but suggest there may be differences in the brain cytokine and metabolic profile in patients who develop more severe disease. Patients who do not respond to therapy are arguably the key group for whom new approaches are urgently needed. Importantly, we have demonstrated that cytokines and metabolites can be tracked in real-time. This can be extended to observe responses to treatment: because MD allows repeated measures over several days, it may be a method to attempt targeted

immunomodulation in the acute phase of the disease in clinical trials, because it enables quantifiable measures of the intervention's effect.

## Limitations

The sample size for this study was small and the results need validation in a larger cohort. Although we examined the temporal profile of cytokines, we were limited to single daily samples because of the small hourly sample volumes and the use of samples for other studies. Higher frequency monitoring is possible and may be useful given that clinical experience in TBM suggests that pathophysiological events such as seizures and infarction can occur rapidly in a patient's time course. Although we elected to examine daily samples and selected time-linked samples, up to 6 samples a day is possible if pooling over a 4-hour epoch was used.

A limitation of MD is the relative recovery of molecules and the difficulty of calibrating the catheter *in vivo*. A further limitation of the technique is the potential difficulty in extrapolating data from the small area of microdialysis monitored brain to the inflammatory state of the entire brain [28]. However, TBM induces a diffuse inflammatory process [29] and the area of MD sampling may be generalisable to the brain as a whole. The important observation is that over the first 4–5 days of monitoring this appears to be a stable methodology that can track the time course of relative concentrations of substances [30]. While it has been suggested that the invasive nature of the MD catheter may itself cause local inflammation that confounds results [31], varying patterns observed in different cerebral pathologies [32, 33] and different patients [34] suggest catheter insertion alone is not responsible for the readings and that after the first hour after insertion the readings are stable. Catheter function over the monitoring time period used in this study is unlikely to be influenced by local disruptions from biofouling, encapsulation or bleeding [11].

This study cohort was biased towards the most severe spectrum of disease, as only very ill patients needed ventilation and intracranial monitoring. Most patients already had infarcts present on their initial scan. Therefore, the data may not generalise to all patients with TBM. Finally, we could not examine for associations with intracranial pressure because patients were undergoing ventricular CSF drainage.

Despite these limitations, this project is the first attempt to analyse inflammatory mediators in brain ECF in patients with TBM, the closest samples to the site of disease *in vivo*, and offers insight into acute cytokine profiles early in treatment, and their immediate response to pathophysiological events. Additionally, while MD may not be used to directly compare cytokine concentrations between compartments or elucidate exact concentrations, relative changes over time can offer valuable temporal evolution data. Further research in a larger cohort, possibly using higher sensitivity analysis methods such mass spectrometry, could elucidate valuable processes occurring in the early, and most critical, part of the disease. This may help develop novel therapies and allow us to track the cerebral response to these.

## Conclusion

This study demonstrated feasibility in detecting inflammatory mediators in brain ECF and their temporal profile in patients with TBM using MD. Despite the limitations, it represents a promising platform to examine disease evolution and possibly the cerebral response to therapies.

## Supporting information

**S1 Fig. Paired brain extracellular, ventricular cerebrospinal and lumbar cerebrospinal fluid cytokine concentrations.** Each LCSF sample analysed (n = 3) is plotted with its paired

ECF and VCSF which were obtained from the same patient at the same time point. Plotted values have been shifted horizontally to avoid overplotting. Abbreviations: ECF, extracellular fluid; IFN-γ, interferon γ; IL, interleukin; IL-1Ra, interleukin 1 receptor antagonist; IP-10, interferon-γ inducible protein-10; LCSF, lumbar cerebrospinal fluid; MCP-1, monocyte chemoattractant protein; MIP-1α, macrophage inflammatory protein 1α; TNF-α, tumour necrosis factor-α; VEGF, vascular endothelial growth factor; VCSF, ventricular cerebrospinal fluid. (TIF)

**S2 Fig. Paired brain extracellular fluid, plasma, and ventricular cerebrospinal fluid inflammatory marker concentrations.** Each plasma sample analysed had both a paired ECF and VCSF which were obtained from the same patient at the same time point (n = 3 cases). Plotted values have been shifted horizontally to avoid overplotting. Abbreviations: ECF, extracellular fluid; IFN-γ, interferon γ; IL, interleukin; IL-1Ra, interleukin 1 receptor antagonist; IP-10, interferon-γ inducible protein-10; LCSF, lumbar cerebrospinal fluid; MCP-1, monocyte chemoattractant protein; MIP-1α, macrophage inflammatory protein 1α; TNF-α, tumour necrosis factor-α; VEGF, vascular endothelial growth factor; VCSF, ventricular cerebrospinal fluid. (TIF)

**S1 Table. Paired brain extracellular fluid and ventricular cerebrospinal fluid cytokine concentrations.**
(DOCX)

**S2 Table. Cytokine differences between radiological outcome and mortality.**
(DOCX)

**S3 Table. Metabolite differences between radiological outcome and mortality.**
(DOCX)

## Acknowledgments

The authors thank the included patients and their guardians. They acknowledge the neurosurgeons at Red Cross War Memorial Children's Hospital for sample collection and the Neurosurgery Research Group who assisted with sample collection, processing, and storage. They also thank the nurses at Red Cross War Memorial Children's Hospital Cape Town for their assistance with the microdialysis monitoring.

## Author Contributions

**Conceptualization:** Anthony Figaji.

**Data curation:** Ursula K. Rohlwink, Mvuwo Tshavhungwe, Lindizwe Dlamini, Muki Shey, Nico Enslin.

**Formal analysis:** Nicholas W. Loxton, Ursula K. Rohlwink, Anthony Figaji.

**Funding acquisition:** Anthony Figaji.

**Investigation:** Nicholas W. Loxton, Ursula K. Rohlwink, Anthony Figaji.

**Methodology:** Ursula K. Rohlwink, Anthony Figaji.

**Project administration:** Ursula K. Rohlwink, Anthony Figaji.

**Resources:** Muki Shey, Anthony Figaji.

**Supervision:** Ursula K. Rohlwink, Anthony Figaji.

**Visualization:** Nicholas W. Loxton, Ursula K. Rohlwink, Anthony Figaji.

**Writing – original draft:** Nicholas W. Loxton, Ursula K. Rohlwink, Anthony Figaji.

**Writing – review & editing:** Nicholas W. Loxton, Ursula K. Rohlwink, Mvuwo Tshavhungwe, Lindizwe Dlamini, Muki Shey, Nico Enslin, Anthony Figaji.

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
