## [Decision Letter · Decision Letter 0]

12 Nov 2020

PONE-D-20-25771

A pilot study of inflammatory mediators in brain extracellular fluid in paediatric TBM

PLOS ONE

Dear Dr. Figaji,

Thank you for submitting your manuscript to PLOS ONE. After careful consideration, we feel that it has merit but does not fully meet PLOS ONE’s publication criteria as it currently stands. Therefore, we invite you to submit a revised version of the manuscript that addresses the points raised during the review process.

Please be sure to respond to the issues raised by Reviewer #2. In particular, can you correlate the findings with measurement of ICP or metabolic factors? 

We look forward to receiving your revised manuscript.

Kind regards,

Richard Bruce Mink

Academic Editor

PLOS ONE

Journal Requirements:

This work is supported by the National Research Foundation SARChI Chair of Clinical

361 Neurosciences grant held by A.F.

362 N.W.L. received financial support from a Harry Crossley Research Fellowship and by

363 the South African Medical Research Council (SAMRC) through its Division of

364 Research Capacity Development under the National Medical Students Research

19

Training Programme from funding received from the South 365 African National Treasury.

366 The content hereof is the sole responsibility of the authors and does not necessarily

367 represent the official views of the SAMRC or the funders.

i) We note that you have provided funding information that is not currently declared in your Funding Statement. However, funding information should not appear in the Acknowledgments section or other areas of your manuscript. We will only publish funding information present in the Funding Statement section of the online submission form.

ii) Please remove any funding-related text from the manuscript and let us know how you would like to update your Funding Statement. Currently, your Funding Statement reads as follows:

 "The author(s) received no specific funding for this work".

iii) Please include your amended statements within your cover letter; we will change the online submission form on your behalf.

Reviewers' comments:

Reviewer's Responses to Questions

**Comments to the Author**

1. Is the manuscript technically sound, and do the data support the conclusions?

Reviewer #1: Yes

Reviewer #2: Partly

2. Has the statistical analysis been performed appropriately and rigorously? 

Reviewer #1: N/A

Reviewer #2: Yes

3. Have the authors made all data underlying the findings in their manuscript fully available?

Reviewer #1: Yes

Reviewer #2: Yes

4. Is the manuscript presented in an intelligible fashion and written in standard English?

Reviewer #1: Yes

Reviewer #2: Yes

5. Review Comments to the Author

Reviewer #1: Very well done pilot study on inflammatory mediators in brain ECF in TBM. It would be interesting to see how inflammatory markers would help in identifying or treating patients with TBM to prevent significant morbidity and mortality. This pilot study might assist patients in developing nations where TB is prevalent and causes significant morbidity and mortality. Having said that techniques like MD are available only in tertiary centers, so co-relation between brain ECF and blood markers can be used as indirect co-markers in underdeveloped nations. Overall, very good study, need to see how markers trend in large set of patient population.

Reviewer #2: The article seeks to explore microdialysis fluid as a source for analysis. This is the main aim of the article. It is written appropriately and tests done well.

It additionally measured cytokines in 7 TBM patients; all severe (1 definite, 6 probable). Note CSF glucose (normal) and protein (slightly elevated) were of uncertain origin (ventricular or lumbar)

The findings are referable to the brain area samples as local stress may cause membrane breakdown with associated cytokine change as eco0rede. As this is microdialysis fluid and focal area of abnormality may occur in TBM such as ischemia, the finding is not generalizable to the whole brain unless in the absence of local pathology and related to a systemic process such as raised ICP or general hypoxia for instance.

Thus the cytokine data is of little clinical value in this study. Longitudinal measurements across may time points over days may yield more useful information, particularly correlated to circulatory, metabolic and ICP measures.

Thus although microdialysis fluid may be used for analysis its value and utility still needs elaboration.

6. PLOS authors have the option to publish the peer review history of their article (what does this mean?). If published, this will include your full peer review and any attached files.

Reviewer #1: **Yes: **Sreenath Meegada MD MPH FACP

Reviewer #2: No

---

## [Author Response · Author response to Decision Letter 0]

27 Dec 2020

We thank the reviewers for the careful assessment of our manuscript and the editorial team for the opportunity to submit a revised version of our manuscript PONE-D-20-25771 A pilot study of inflammatory mediators in brain extracellular fluid in paediatric TBM. We appreciate the supportive comments and below we address the issues raised.

Response to editorial comments:

1. The manuscript has been formatted as per PLOS ONE requirements.

2. Once the manuscript has been formally accepted we will deposit the data in ZivaHub and provide the reference details.

3. We have removed any funding statements from the manuscript, please kindly update our funding statement to include: “This work is supported by the National Research Foundation SARChI Chair of Clinical Neurosciences grant held by A.F.

N.W.L. received financial support from a Harry Crossley Research Fellowship and by the South African Medical Research Council (SAMRC) through its Division of Research Capacity Development under the National Medical Students Research Training Programme from funding received from the South African National Treasury. The content hereof is the sole responsibility of the authors and does not necessarily represent the official views of the SAMRC or the funders.” 

Response to Reviewers' comments:

Reviewer #1: Very well done pilot study on inflammatory mediators in brain ECF in TBM. It would be interesting to see how inflammatory markers would help in identifying or treating patients with TBM to prevent significant morbidity and mortality. This pilot study might assist patients in developing nations where TB is prevalent and causes significant morbidity and mortality. Having said that techniques like MD are available only in tertiary centers, so co-relation between brain ECF and blood markers can be used as indirect co-markers in underdeveloped nations. Overall, very good study, need to see how markers trend in large set of patient population.

Response: We thank the reviewer for their complimentary assessment of our manuscript. We agree that MD is unlikely to be feasible in underdeveloped nations, and we do not intend to promote its use for this purpose. Our aim is to develop pilot data to determine whether MD is feasible and could be a tool which, in addition to having clinical value, also enables the real-time longitudinal study of the brain to examine its response to clinical interventions, especially in advanced centers and in clinical trials. If this is successful we can propose possible underlying mechanisms of action and support the development of clinical trials of specific interventions. We agree that further investigation in a larger cohort will be important in determining its use in this context.

Reviewer #2: The article seeks to explore microdialysis fluid as a source for analysis. This is the main aim of the article. It is written appropriately and tests done well.

It additionally measured cytokines in 7 TBM patients; all severe (1 definite, 6 probable). Note CSF glucose (normal) and protein (slightly elevated) were of uncertain origin (ventricular or lumbar).

Response: We thank the reviewer for their comments. We would like to draw their attention to line 142 of the Results section and to Table 1 in which we state that ventricular CSF data are presented due to limited availability of lumbar CSF for this cohort of patients. These patients alll were in the intensive care unit and had ventricular CSF drainage. We highlight this because of the relatively mild CSF changes as opposed to typical lumbar CSF findings.

The findings are referable to the brain area samples as local stress may cause membrane breakdown with associated cytokine change as eco0rede. As this is microdialysis fluid and focal area of abnormality may occur in TBM such as ischemia, the finding is not generalizable to the whole brain unless in the absence of local pathology and related to a systemic process such as raised ICP or general hypoxia for instance.

Thus the cytokine data is of little clinical value in this study. Longitudinal measurements across may time points over days may yield more useful information, particularly correlated to circulatory, metabolic and ICP measures.

Thus although microdialysis fluid may be used for analysis its value and utility still needs elaboration.

Response: We thank the reviewer for the insightful comment. He/she is correct in stating that microdialysis – like any focal monitoring tool – measures a limited area of the brain; therefore the results may not be generalizable to other areas of the brain. We acknowledge as much in the limitations sections of our manuscript. Based on our experience though – and the global community of specialists involved in advanced neurocritical care – focal monitoring often does detect insults that the whole brain is subjected to. This is our experience also with brain tissue oxygen monitoring, with which we now have experience in almost 300 patients and which mirrors our experience with TBM. The hypothesis is that even though certain focal areas develop infarction, the insult to the brain is often much more widespread (because it is due to increased ICP, hypoxia, or hypotension), and the areas that eventually infarct tend to be in perforator territories that are most vulnerable. This is also consistent with our observation over several years of brain tissue oxygen monitoring in TBM patients where traditional intracranial pressure thresholds may not be applicable because of the underlying vasculitis (being prepared for publication). In these patients, general measures such as intracranial pressure reduction, increased blood pressure and increased systemic oxygenation result are demonstrate in the focal region of examination.

We agree fully though, that these are complicated questions and hypotheses, and the clinical and research role of these tools still require elucidation. This is precisely what we are aiming to do, and this manuscript is part of that process. 

We also agree with the reviewer’s comment that the longitudinal examination of these changes over time may yield better information than single values in absolute terms, which we address in the Discussion.

The reviewer specifically mentions the possible association with other measures, such as intracranial pressure and metabolites. We did not do this anlaysis as part of the original manuscript because the patient cohort was small and any statistical analysis would meaningless given the need to adjust for repeated measures. However, in response to the question we have performed a preliminary hypothesis-generating analysis of the cytokines and metabolites in association with radiological and clinical outcome. We could not analyse against intracranial pressure because these patients had ventricular drainage set to normalize the pressure. 

We proceeded to analyse radiological features, clinical outcome and metabolites. We reviewed the radiological findings from formal reports of brain computed tomography (CT) scans. In our previous work, we demonstrated that simple classification according to the presence or absence of infarcts alone is not as predictive as might be expected, possibly because infarcts vary substantially in size and location, both of which have important implications for death or disability. Also, 6 of the 7 patients developed infarcts. So we classified patients into two groups based on the characteristics of infarcts: Group A comprised patients with either focal infarcts or no infarcts; Group B comprised patients with extensive hemispheric or bilateral infarcts.

Because the patient cohort was so small and the analysis was preliminary, we felt that group statistics would be unreliable. Any attempt to do so would be meaningless given the small size and the necessary adjustment for repeated measures testing. To generate hypotheses, we highlighted differences in brain cytokines or metabolites in which there was at least a 3-fold difference between groups.

Clinical outcome was mortality at one year follow up.

Metabolites were examined from hourly analysis on the bedside Mdialysis analyzer.

Infarct Group A and B

There were 5 patients in Group A and two patients in Group B (extensive and/ or bilateral infarction).

For cytokines:

1) Median cytokine values in Group B were greater than or equal to corresponding values in Group A for all cytokines.

2) Cytokines that demonstrated median values at least 3-fold greater: IL-1β, IL-1Ra, IL-6, and MCP-1 (in Group B).

For metabolites: 

Median glycerol values were 3-fold greater in Group B (145.79 versus 40.28); the lactate-pyruvate ratio was slightly higher (24.24 versus 19.51), and glucose was slightly lower (0.61 versus 1.19).

Clinical Outcome

For clinical outcome, patients were classified as dead (3 patients) or alive (4 patients). All survivours achieved a good functional outcome at one year follow up.

For cytokines:

1) Median cytokine values in patients who died were greater than or equal to corresponding values in survivours for all cytokines, except Il6. 

2) Cytokines that demonstrated median values at least 3-fold greater: Il1B and MCP1 (in patients who died).

For metabolites:

Median glycerol values were 3 fold greater in patients who died (76.03 versus 24.62); the lactate-pyruvate ratio was slightly higher (23.43 versus 19.23), glucose was slightly lower (0.93 versus 1.14).

We have added the relevant text above to the Methods and Results sections (see track changes) along with Tables S2 and S3.

We have added the following to the Discussion

‘Although the sample size is small, the data suggest some intriguing hypotheses that require examination in a larger cohort. In general, cytokine values were greater in patients who died, and who experienced extensive infarction. In particular, Il1B, Il1Ra, Il6, and MCP1 showed 3-fold higher values in patients with extensive infarction; for patients who died, Il1B and MCP1 were 3-fold greater. 

Metabolites showed less marked differences, but glycerol was 3-fold greater in patients who died and patients who developed extensive or bilateral infarction, which may be in keeping with tissue destruction. 

These observations require validation in a larger cohort, but suggest there may be differences in the brain cytokine and metabolic profile in patients who develop more severe disease. Patients who do not respond to therapy are arguably the key group for whom new approaches are urgently needed. Importantly, we have demonstrated that cytokines and metabolites can be tracked in real-time. This can be extended to observe responses to treatment: because MD allows repeated measures over several days, it may be a method to attempt targeted immunomodulation in the acute phase of the disease, because it enables quantifiable measures of the intervention’s effect.’

Once again, we thank the editor and reviewers for their comments and opportunity to strengthen the manuscript. We hope that it is now acceptable for publication.

---

## [Decision Letter · Decision Letter 1]

1 Feb 2021

A pilot study of inflammatory mediators in brain extracellular fluid in paediatric TBM

PONE-D-20-25771R1

Dear Dr. Figaji,

We’re pleased to inform you that your manuscript has been judged scientifically suitable for publication and will be formally accepted for publication once it meets all outstanding technical requirements.

Kind regards,

Richard Bruce Mink

Academic Editor

PLOS ONE

Additional Editor Comments (optional):

Reviewers' comments:

Reviewer's Responses to Questions

**Comments to the Author**

1. If the authors have adequately addressed your comments raised in a previous round of review and you feel that this manuscript is now acceptable for publication, you may indicate that here to bypass the “Comments to the Author” section, enter your conflict of interest statement in the “Confidential to Editor” section, and submit your "Accept" recommendation.

Reviewer #2: All comments have been addressed

2. Is the manuscript technically sound, and do the data support the conclusions?

Reviewer #2: Yes

3. Has the statistical analysis been performed appropriately and rigorously? 

Reviewer #2: Yes

4. Have the authors made all data underlying the findings in their manuscript fully available?

Reviewer #2: Yes

5. Is the manuscript presented in an intelligible fashion and written in standard English?

Reviewer #2: Yes

6. Review Comments to the Author

Reviewer #2: The authours have addressed commnets and adjusted teh limitations to reflect teh value ofteh article. The conclusion appropriately limits itself to the technique suggeted.

7. PLOS authors have the option to publish the peer review history of their article (what does this mean?). If published, this will include your full peer review and any attached files.

Reviewer #2: No

---

## [Editor Report · Acceptance letter]

4 Mar 2021

PONE-D-20-25771R1 

A pilot study of inflammatory mediators in brain extracellular fluid in paediatric TBM 

Dear Dr. Figaji:

I'm pleased to inform you that your manuscript has been deemed suitable for publication in PLOS ONE. Congratulations! Your manuscript is now with our production department. 

Kind regards, 

on behalf of

Dr. Richard Bruce Mink 

Academic Editor

PLOS ONE